# Proto-Early Renaissance Depictions, Iconographic Analysis and Computerised Facial Similarity Assessment Connections: The 16th Century Mural Paintings of St. Leocadia Church (Chaves, North of Portugal)

**Eunice Salavessa** [1,*] , **José Aranha** [2] , **Rafael Moreira** [3] **and David M. Freire-Lista** [4,5]

1 Forestry Sciences and Landscape Architecture Department, University of Trás-os-Montes and Alto Douro, 5001-801 Vila Real, Portugal

2 Departamento de Ciências Florestais e Arquitetura Paisagista, Centro de Investigação e Tecnologias Agroambientais e Biológicas, Universidade de Trás-os-Montes e Alto Douro, 5001-801 Vila Real, Portugal; j_aranha@utad.pt

3 CHAM, Centro de Humanidades, Faculdade de Ciências Sociais e Humanas, Universidade Nova de Lisboa, 1069-061 Lisboa, Portugal; rfdmoreira@gmail.com

4 Departamento de Geologia, Universidade de Trás-os-Montes e Alto Douro, 5001-801 Vila Real, Portugal; davidfreire@utad.pt

5 Centro de Geociências da Universidade de Coimbra, Universidade de Coimbra-Polo II, 3030-790 Coimbra, Portugal

* Correspondence: eunicesalavessa@sapo.pt or e_salave@utad.pt

**Abstract:** The aim of this paper is to analyse facial similarity and apply it to identify the individuals depicted in the mural paintings of the apse of St. Leocadia Church, located in Chaves Municipality (North of Portugal), which were painted during the first quarter of the 16th century. This study also compares the portraits of this mural paintings with the oil paintings by the Proto-Renaissance Portuguese painter Nuno Gonçalves. Through this research, the feasibility of face recognition technology is explored to answer many ambiguities about Manueline stylistic identity and iconography. Additionally, it aims to associate historical events, artistic discoveries, and the expansion of portraiture as propaganda of power during the Portuguese Proto-Renaissance and Early Renaissance. On the other hand, it focuses on the prevalence of the religious and devotional over the sacred in Manueline painting. A proposal was made to identify the characters that are fundamental to the meaning of the mural paintings. An experiment was conducted on seven characters from the paintings at St. Leocadia Church, which were then compared to Nuno Gonçalves' portraits. Facial similarity analysis was conducted on the faces portrayed in the Panels of St. Vincent, a remarkable portrait gallery from 15th-century Portugal, which has been the subject of national and international research for 130 years. Other paintings that were analysed were the oil paintings of St. Peter and St. Paul and of Infanta St. Joana, which were created by the same Quattrocento master. The purpose of the mural paintings of St. Leocadia Church could be catechetical in nature or related to the ritual practices of royal ancestor worship in royal portrait apses of the churches. It could also be associated with the Portuguese maritime expansion and the macro-imperial ideology of D. Manuel I.

**Keywords:** deep learning; face recognition; Early Renaissance mural paintings; Proto-Renaissance oil paintings; style modelling

## 1. Introduction

Creating a comprehensive database of various analytical techniques is crucial when selecting the most appropriate solutions for restoring mural paintings. The investigation methodology concerning works of art involves identifying and resolving stylistic problems, as well as understanding the tools and techniques required to reconstruct damaged or missing components [1–6]. The facial resemblance in portraits provides valuable information

for questions related to the identity of the portrayed individuals and their historic-artistic significance. It helps us understand why a certain person was painted in a specific manner. This study focuses on exploring portrait artworks from the Proto-Renaissance and Early Renaissance periods, such as the Vicentine panels and other oil paintings by Nuno Gonçalves, and mural paintings of the apse of St. Leocadia Church. Advanced face recognition technologies are utilised to identify the represented characters in the mural paintings and reconstruct the organization and layout of the images. In this way, more meaning is brought to the narrative presented. Moreover, it highlights the Manueline iconographic program as a common characteristic of these artworks [7,8]. St. Leocadia Church is near a small namesake village located just a few miles south of defensive walls of Chaves town. It has a longitudinal plane, a nave, and an apse, and its origins are Romanesque. In the 12th century, it served as the parish seat of the Jurisdiction of Montenegro. The church is mentioned in the 13th and 14th centuries as a rectory and commendation of the landlord of Bragança [9,10]. During the first quarter of the 16th century, the interior of St. Leocadia Church was decorated for symbolic reasons related to the political and religious beliefs of the time. The decoration was based on what was important to King D. Manuel I and the parishioners of St. Leocadia village, which included their maritime achievements during his reign (Figures 1 and 2). The objective of this decoration was to dignify the religious space and to recover the prestige of the Duchy of Bragança. This church is a Public Interest Site (Decree-Law nº. 44 075, DG, 1st series, nº. 261, 5 December 1961). Between 1996 and 2003, architectural heritage organisations (DGEMN and EPRPS) worked on analysing, conserving, and restoring the interior walls and mural paintings of the nave and apse. Similarities can be found in a series of mural paintings from the same period, in the Vila Real district. Such as, themes, ornamental elements, iconographic influences and programme correlations between the mural paintings in the Santa Leocádia Church and those in the St Marinha Church in Vila Marim, the Our Lady Chapel of Guadalupe in Mouçós, the St. Michael Church of Três Minas and the Our Lady Church of Azinheira in Outeiro Seco [11–15].

The medieval stonework of the apse is now covered by mural paintings in the form of triptychs. On the back wall, there are represented the Apostles St. Peter and St. Paul, each located on the opposite sides of the St. Leocadia statue. On the side walls, there are several depictions of moments in the lives of Our Lady and Jesus until his adolescence: The Visitation, The Annunciation to the Shepherds, The Presentation of the Child Jesus in the Temple, the Flight into Egypt, The Slaughter of the Innocents, and Jesus among the Doctors (Figure 2). Those paintings are displayed in "memory niches" or observation balconies. They are situated on a low wall that is covered with a painting that imitates quadrangular modules in a Renaissance diamond point. The greyish stone columns and architraves of the "loggie" frame the mural paintings, are arranged in a series of spatial cubes that project into depth, creating a modular space. The architectural setting is topped by a red and grey frieze of profane fantasy figures. In this painting, there are several mythical and fantastical creatures such as centaurs, mermaids, griffins, and beings with winged heads. One of the figures is riding a horse and holding a spear, while a centaur is depicted throwing a sabre. The painting also includes Florentine amphoras, branches with flowers and ivy leaves, as well as various animals like herons, falcons, orioles, doves, greyhounds, and wild boars. The decoration is charming, painted in silver on a scarlet velvet background. They could be related to Raphael's paintings in the Patio de San Damaso, in the Vatican Palace, inspired by the grotesque decoration of the Domus Aurea. It can recall several magnificent works of art, such as the sculptural frieze created by João de Castilho, which portrays exotic sea-inspired designs and can be found at the Convent of Christ; the bicoloured Indian bedspreads from the 16th century; reminiscent of the red velvets embroidered with silver silk from the Trás-os-Montes region in the 16th century; and the Indo-Portuguese Cinquecento boxes, which are made of red lacquered wood with silver decoration and are based on Greco-Roman mythology [16–18].

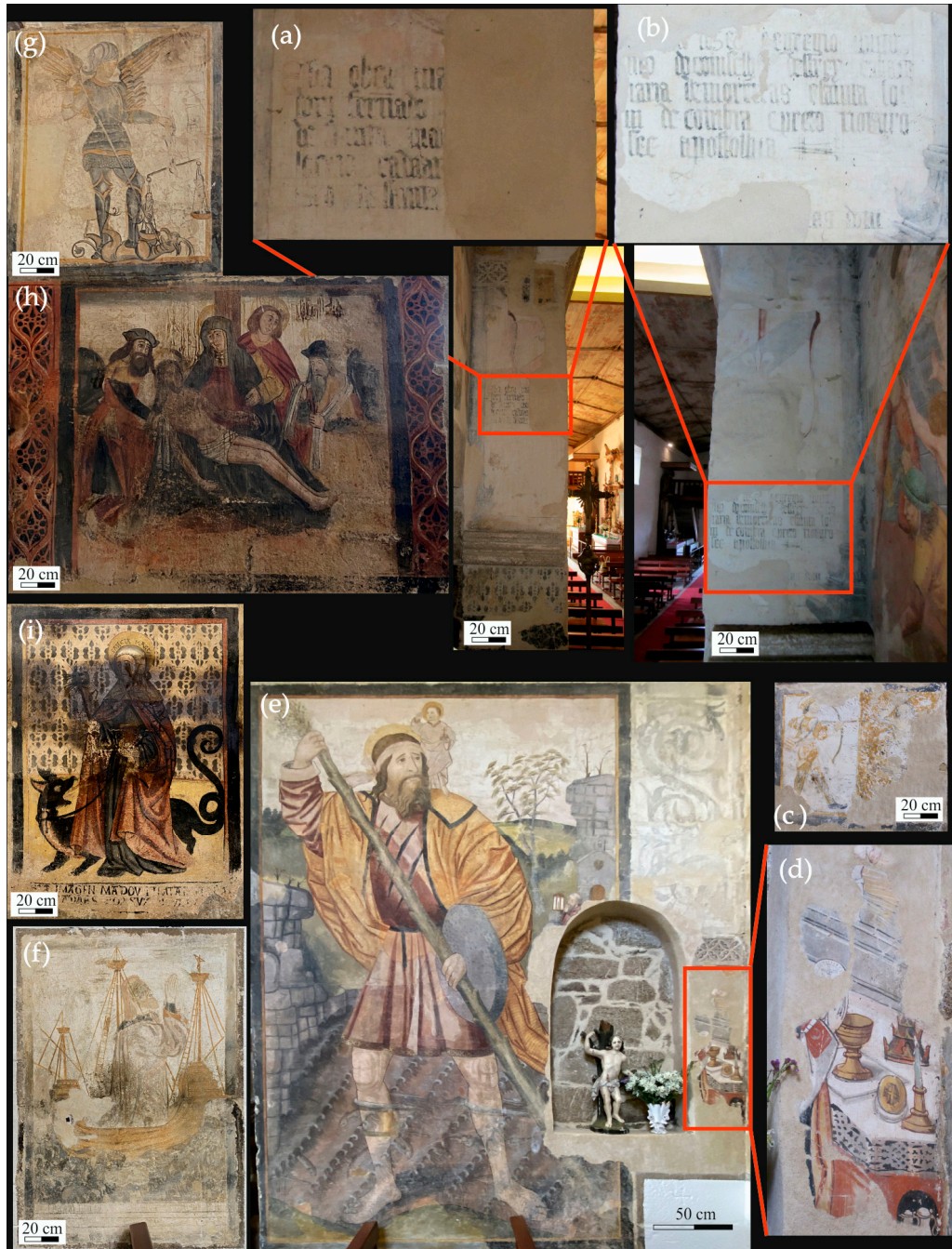

**Figure 1.** Mural paintings of St. Leocadia Church: (**a**,**b**) bipartite composition of an inscription, in the Epistle and Gospel sides of the triumphal arch of the apse. In the nave: (**c**) Martyrdom of St. Sebastian; (**d**) St. Gregory's Mass; (**e**) St. Christobal; (**f**) Caravel (ex-voto); (**g**) Archangel St. Michael; (**h**) Lamentation over the Dead Christ; and (**i**) St. Martha.

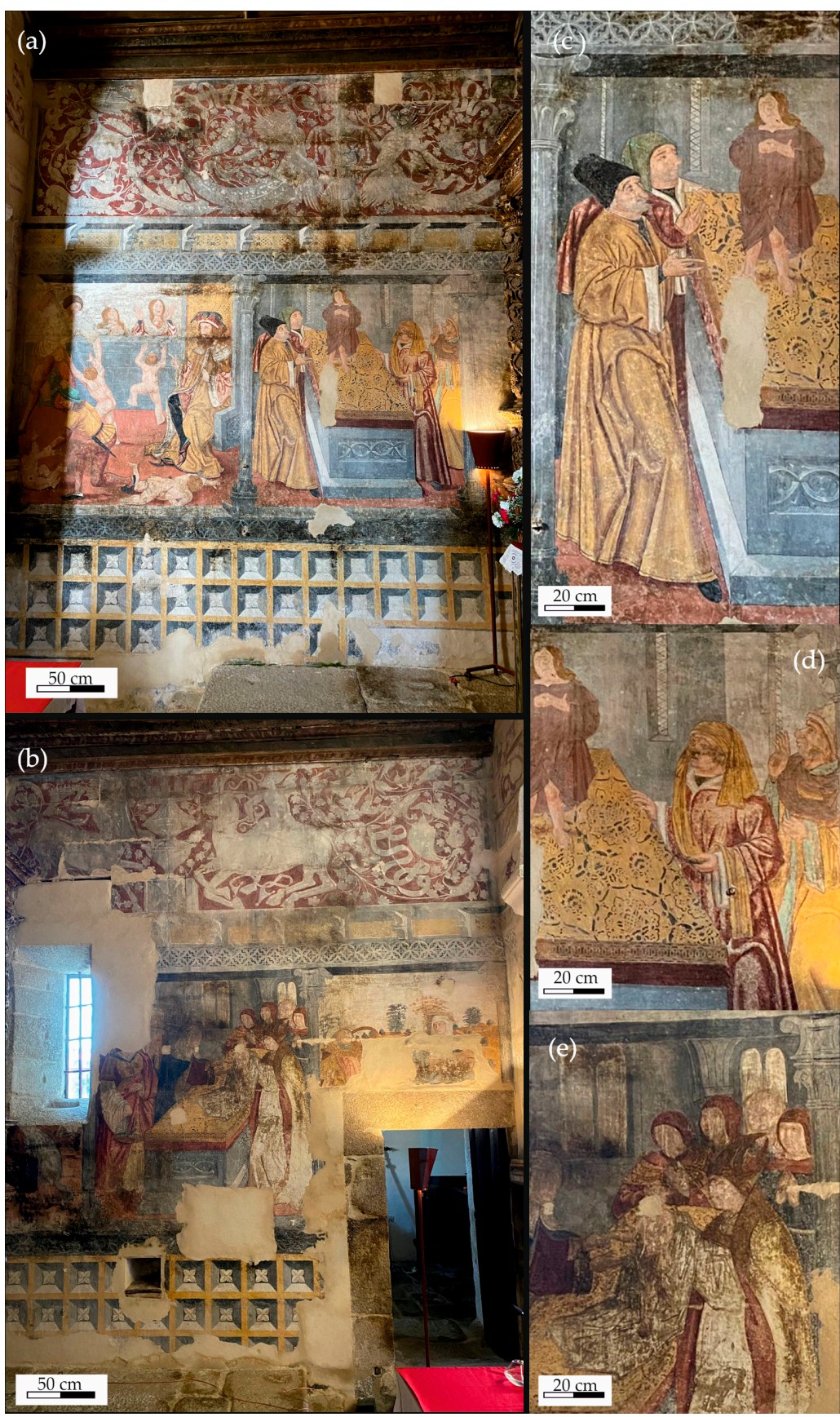

**Figure 2.** Mural paintings of the apse of St. Leocadia Church: (**a**) on the Gospel wall, The Slaughter of the Innocents (175 × 175 cm) and Jesus among the Doctors (175 × 175 cm); (**b**) on the Epistle wall,

Jesus' Presentation in the Temple (175 × 175 cm) and The Flight into Egypt (175 × 92 cm); a comparative visual analysis is proposed between the characters in the mural paintings of St. Leocadia Church and those in the Vicentine panels: (**c**) Estevão de Aguiar and Jewish rabbi, of the Vicentine Panels of the Friars and of the Relic; (**d**) Friar João Alvares of the Panel of the Relic; and (**e**) Archbishop, D. Alvaro Vaz de Almada, D. Jaime, and D. Pedro Contestable, of the Vicentine Panel of the Archbishop.

The Ethical Renaissance is revealed in the apse of St. Leocadia Church. The paintings follow the principles of representing the "ethical body", typical of Alberti's aesthetic thinking. Leon Battista Alberti, in the second book of "De Pictura" (1435) suggests that bodies should be represented according to the "istoria" or events with which they are involved. It is not the physical beauty of the bodies or their naturalistic representation, the great achievement of the painter, but the beauty of "istoria" and the transmission of facts and ethical values to the viewer. Bodies must be expressed themselves in a historical context. Representing "istoria" has an ethical and pedagogical meaning, conforming souls within a fairer and better life that men should endeavour to build in their public and private lives. Alberti also said of posthumous painting: "It contains the divine power to make the absent present; even more so, to make the dead, after many centuries, almost living beings (...)" [19]. In the mural paintings of St. Leocadia Church, we find the history of the Avis dynasty incorporated into the sacred scenes themselves [20,21], with the intention of not only perpetuating the memory of virtue and valour as a Renaissance "ethos", but also of legitimising the dynastic family continuity in power.

## 2. Literature Review on Facial Similarity Assessment

On the need to construct a frame of verticals and horizontals that intersect the visual rays in perspective, Leon Battista Alberti said: "This cut of the veil, in fact, certainly offers not few opportunities, first because it always presents the same surfaces unchanged. (...) how it is impossible to correctly imitate what does not, without interruption, maintain the same aspect. That's why it's easier to portray painted things than sculpted ones. We know that by changing the distance and the position of the vertex of the visual pyramid, what we see seems very altered. Therefore, the veil will be extremely useful to us because, when we see something, it will always be the same." On examining the proportions of the members: "If in a painting the head were very large, the chest small, the hand broad, the foot swollen and the body turgid, the composition would certainly be deformed to sight. That's why it's important to keep a certain ratio behind the size of the limbs. When dimensioning a living being, it will be useful to place, first, each bone of the living being, then add its muscles, then will cover it with its flesh. (...) In order to dimension correctly a living being, you must select a member of the same being, with which you will measure all others." [19].

The latest advancements in face recognition systems have made a significant impact on the analysis of portrait works. These systems provide quantitative information to art historians, making it easier for them to identify and compare the facial similarities of the characters depicted in the portraits. Srinivasan et al. 2013, explored the potential of face recognition technologies in answering many of the ambiguities concerning the identity of the subject in some portraits and in understanding artists' styles. Portrait Feature Space (PFS) was utilized to measure the similarities between pairs of Renaissance portraits that are known to represent the same or different subjects. Using statistical hypothesis tests, the researchers analysed portraits where the identities were uncertain and explained the significance of the results from an art historian's perspective [22]. Gupta et al. 2018, made significant improvements over baselines and state-of-the-art methods on various examples that were identified by art historians as being very controversial. To measure the similarity between a pair of portraits, deep Siamese convolutional neural networks (CNNs) were used. To overcome limited training data, a CNN-based style-transfer technique was used to create new images by applying an art style to other images, while keeping the original image content unchanged [23]. Two research projects were

conducted on the topic of sketch recognition, in 2019. Chokkadi et al. have identified sketch recognition as one of the most important areas that have evolved as an integral component, adopted by many law enforcement agencies. The research is based on CNNs and focuses on deep learning techniques used in face recognition and matching. It includes the automatic composite sketch recognition technique, among others [24]. Elmahmudi and Ugail conducted experiments to investigate the effectiveness of face recognition using partial facial data. They tested the performance of machine learning by manipulating face images through techniques like rotation and zooming, which served as training and recognition cues. They studied how the recognition rate varied based on different parts of the face, such as the eyes, mouth, nose, and cheeks. In the research, face recognition was studied concerning facial rotation and zooming out of facial images. It was utilized state-of-the-art CNN architecture and a pre-trained VGG-Face model to extract features for machine learning. Experiments were conducted on two publicly available datasets, namely, the controlled Brazilian FEI and the uncontrolled LFW dataset. According to this study, recognition rates for individual facial features like eyes, nose, and cheeks are low. However, the recognition rate increases significantly when these individual parts are presented together as probes [25]. Hsiao et al. 2021, suggested that part-based information is crucial for face recognition along with whole-face information. In the study, researchers used Eye Movement Analysis with Hidden Markov Models (EMHMM) to analyse face recognition. The findings showed that artists have an edge in face processing, particularly for tasks that are similar to their drawing experience, such as face matching. This advantage may be due to their ability to identify information that remains consistent across different faces, rather than focusing on patterns of eye movement. The results also suggest that artists' skills in face processing are specific to certain tasks and are linked to their drawing experience [26]. Ugail et al. 2022, used an in-house trained deep face recognition model to extract facial features from portraits of people from the 16th and 17th centuries. They compared the degree of similarity between the portraits by using the VGG deep learning model as a basis. The analysis paid particular attention to prominent parts of the face, such as the eyes, nose, and mouth features to enhance facial similarity analysis [27]. Marco et al. 2022, utilized face recognition (FR) technology to enhance the reliability of important decisions. They proposed a model that aimed to improve the transparency of verification decisions. Their approach involved estimating the uncertainty of face comparison scores and introducing a confidence measure of the system's decision. This would provide valuable insights into the verification process. The effectiveness of this model was tested on three face recognition models using two datasets [28]. Hangaragi et al. 2023, have been working on research to develop applications that can identify people in specific areas like stores and banks, control access to restricted areas, ATMs, or computers, and recognize individuals in police databases. They proposed a model that uses face mesh technology to detect and recognize similar faces accurately. The model is capable of processing images of males and females of any age and ethnicity, even if they are not facing the camera directly. The model's deep neural network has been trained using images from the Labeled Wild Face (LWF) dataset and real-time captured images. By comparing the face landmarks of a test image with the landmarks of the training images, the model can identify the person in the image with almost perfect accuracy. The proposed model has achieved high accuracy for face recognition, as mentioned in reference [29]. Grace Zhong 2023, presented a paper that examined the similarities between faces in portraiture. This study can be particularly useful for art historians who want to determine the identity of the sitter, as well as gain insights into the historical context of a painting. The portraits studied were from the Song dynasty in China and were mainly of royalty. A research paper demonstrated the usefulness of computer vision-based quantitative metrics in complementing existing subjective evaluations. The study used the portrait set and L2 distances generated using OpenFace support to test the hypothesis that two emperors of the Song dynasty were identified in surrounding posthumous portraiture, despite confounding factors in the clouds of memory. The results of this research show the potential of using computer vision-

based techniques as complements to subjective analyses in exploring old painted portrait faces [30].

## 3. Materials and Methods

A rigorous working method was followed, utilizing logical, deductive, and inductive reasoning, while avoiding arbitrary character and symbol identifications.

### 3.1. Iconographic Analysis

While researching the conservation of Romanesque churches in the Trás-os-Montes region, unexpected issues arose regarding the interpretation of the iconography. The researchers utilized new artistic discoveries that were previously unknown. This study highlighted the importance of examining the 16th century mural paintings discovered in St. Leocadia Church. One of the primary objectives of the research was to establish a correlation between the paintings most usually attributed to Nuno Gonçalves (active between 1450 and 1472), as the famous Panels of St. Vincent, the oil paintings of St. Peter and St. Paul and the portrait of Princess St. Joan and the main characters represented in the mentioned mural paintings. In addition to the biblical connotation, specific features of Manueline interventions were also identified. One important aspect that required more detailed analysis was the significant role played by symbols and icons in the storytelling of the Manueline style. First, a bibliographical study was conducted where various documents were collected. Documentary and bibliographical collection, original documents and information were obtained from the National and Municipal Archives and Libraries, the National Museum of Ancient Art (MNAA) and the Library of Art History Department, in the NOVA University of Lisbon, as well as published works on the subject by authors and organisations of recognised scientific standing [31,32]. This study is largely iconographic, and so, was carried out a comprehensive photographic survey of the mural paintings of St. Leocadia Church, research on Medieval literature and Renaissance philosophy, history of theology and history and treatises of art, as well as manuscript illuminations and images. The research conducted a thorough analysis of the mural paintings, including their themes, styles, and chronology, intending to gain a deeper understanding of the patron who commissioned the paintings in the apse. Additionally, the artistic trends of the first Renaissance period in Europe were examined, to better contextualize the artistic work.

### 3.2. Deep Face Recognition Model

The deep face recognition model used in this project (Figure 3) was based on the model VGG, which uses a CNN architecture based on deep learning. For this work, an in-house-trained deep face recognition model was used, similar to the one proposed by Ugail, Edwards, Benoy, and Brooke in their research paper [18]. Specifically, the research examined the similarities between the faces in the oil paintings of seven characters from the Vicentine panels and other paintings by the same artist and those depicted in the mural paintings of the apse of the church. There were used seven pairs of images (as shown in Figure 4) to train and test facial feature models for face recognition. The face recognition model was based on Computer Aid Design (CAD) techniques which enabled the assigning of the photo pairs and the measurement of the face features as below:

(a) All the photographs used were recorded using a digital camera.
(b) All photographs were then processed with an image software to create pairs of similar faces to be analysed.
(c) The photograph pairs were then resized so that the faces had the same size and proportions.
(d) Using CAD software (https://www.autodesk.com.au/solutions/cad-software, accessed on 10 February 2024), two rectangles, with the approximate dimensions of the two faces (pairs of images), were created to enable measuring procedures on each photograph.

(e) Several vectors were created over each photograph representing the anatomical features (Figure 3):

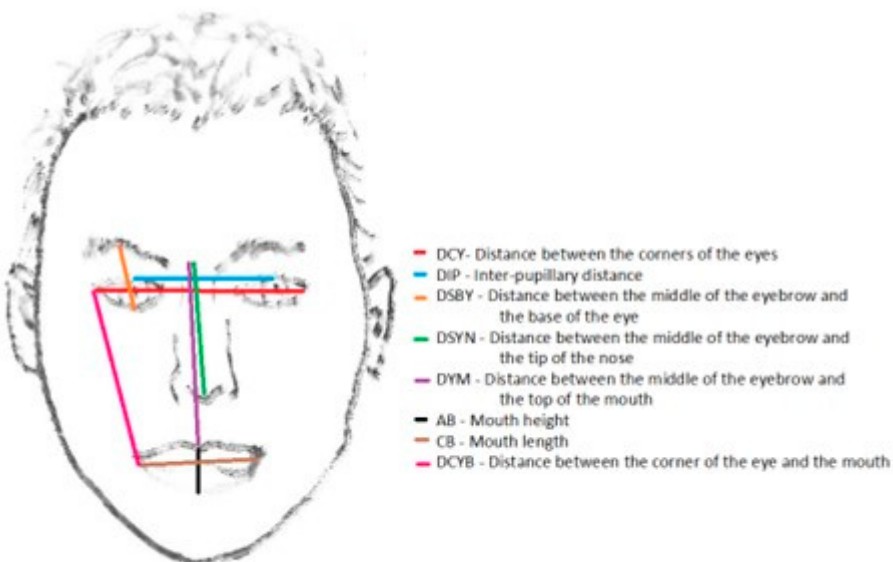

**Figure 3.** Schematic representation of the created vectors.

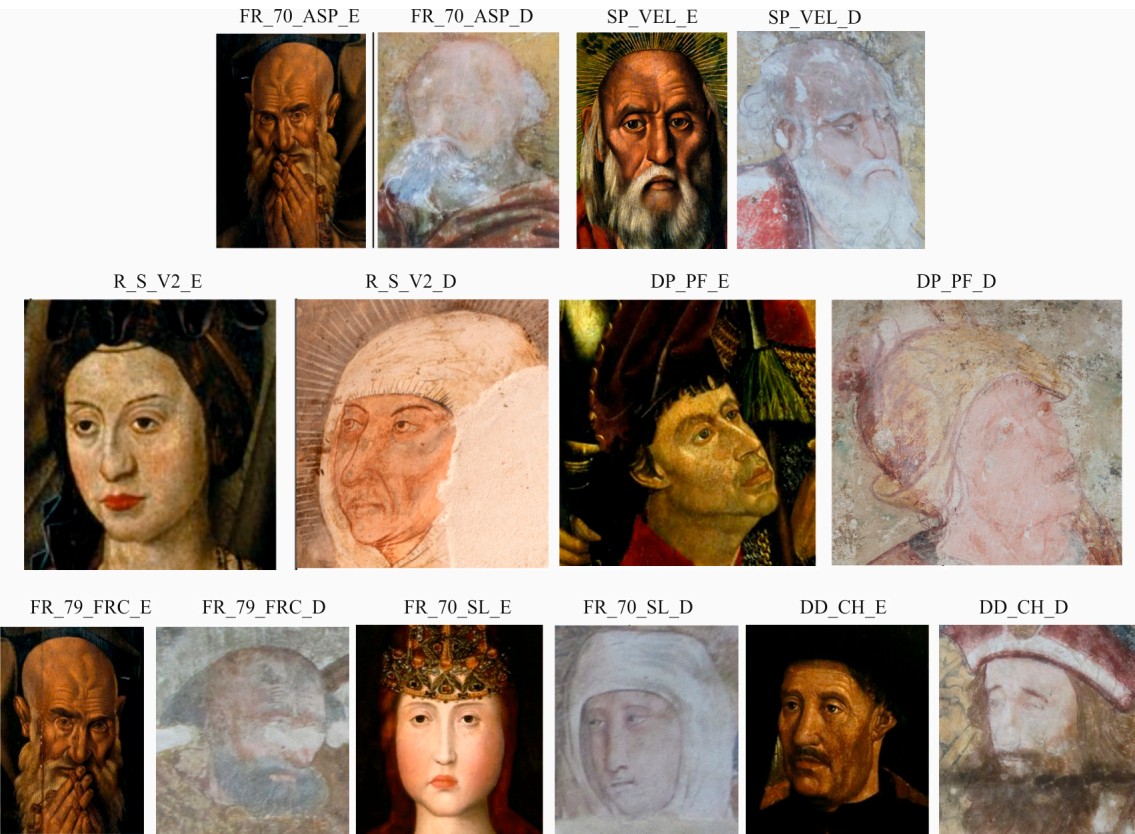

**Figure 4.** Selected images for the training and testing of facial feature-specific models. FR-70_ASP_E vs. D: Franciscan friar vs. St. Peter; SP_VEL_E vs. D: St. Paul vs. St. Paul; R_SV2_E vs. D: Queen Elizabeth vs. St. Elizabeth of the Visitation; DP_PF_E vs. D: Infante D. Pedro vs. the Shepherd of the Annunciation; FR_79_FRC_E vs. D: Franciscan friar vs. St. Joseph of the Flight into Egypt; FR_70_SL_E vs. D: Princess St. Joan vs. Our Lady of the Flight into Egypt; and DD_CH_E vs. D: King D. Duarte vs. the man with a chaperon of the Slaughter of the Innocents.

DCY—Distance between the corners of the eyes;
DIP—Inter-pupillary distance;
DSBY—Distance between the middle of the eyebrow and the base of the eye;
DSYN—Distance between the middle of the eyebrow and the tip of the nose;
DYM—Distance between the middle of the eyebrow and the top of the mouth;
AB—Mouth height;
CB—Mouth length;
DCYB—Distance between the corner of the eye and the mouth.

(f) The previously presented vectors were drawn over each photograph in each pair. The achieved results (for each pair) were used to create an Excel file to be used in further analysis. Based on previously presented variables, each Excel file was updated with the anatomical distances' ratios:

- Ratio between the distance between the corners of the eyes and the distance between the middle of the eyebrow and the tip of the nose;
- Ratio between the distance between the corners of the eyes and the distance between the middle of the eyebrow and the top of the mouth;
- Ratio between the distance between the corners of the eyes and the distance between the corner of the eye and the corner of the mouth;
- Ratio between the distance between the corner of the eye and the corner of the mouth and the length of the mouth;
- Ratio between the distance between the middle of the eyebrow and the base of the eye and the inter-pupillary distance.

(g) An XY representation of the points measured for each pair of images was created, using X for the image of the Panels of St. Vincent and Y for the image of the church mural paintings.

(h) A linear regression equation without a constant was fitted to each pair of values (pairs of images).

(i) The coefficient of determination associated with each equation was calculated.

## 4. Results

New information about the Panels of St. Vincent has been discovered. This information has allowed the creation of a database that includes the author of the polyptych, the date it was executed, where the work was intended to be displayed, and some previously unknown details about the characters depicted in the panels. Through studying the paintings of St. Leocadia Church, we can learn about the religious and philosophical beliefs in Portugal during the early 16th century. The images, stories, and allegories present in the paintings are symbolic representations of these underlying principles. By identifying similarities between facial features in old portraits and mural paintings, the study aimed to offer an interpretation of the historical figures. It should be noted that this task is not easy, and the results cannot be considered definitive. Instead, this study offers only conjectures.

### 4.1. Background

4.1.1. Historical–Political–Artistic Framework

D. João I ushered in a new era in the 15th century, which was marked by an illustrious generation. After his reign, Infante D. Pedro took over and his regency period saw a fracture in Portuguese society. The period culminated in the Battle of Alfarrobeira in 1449. During this time, a "new" nobility emerged with the intention of asserting itself. They defeated the centralizing policies of the real power in this battle and established themselves as a dominant force in society [33]. During the reign of King D. Afonso V, Portuguese art reached its peak with the work of Nuno Gonçalves, the royal painter. The roots of this art can be traced back to the North of Portugal and Galicia (Spain). Despite being influenced by foreign styles, the Portuguese school of painting was characterized by an exceptional representation of the human figure and psychological portraiture, as well as

technical mastery. This school of painting existed before the Flemish influence on Portugal, which occurred after Jan van Eyck arrived in the country in 1428. The Panels of St. Vincent, painted by Nuno Gonçalves, are considered to be on a par with the artistic level of Spain, France, Italy, and Flanders [34].

During the reigns of D. João II and D. Manuel, Portugal experienced a period of great expansion and growth. The country's relationship with other kingdoms in the Iberian Peninsula was strengthened, and this era is known as the Portuguese Golden Age. D. Manuel introduced significant changes to governance, restructuring commerce and navigation (with the establishment of the House of India being a notable achievement). The country's power was influenced by a new vision of expansion and missionary work. Notably, the Church also changed during this period, with administrative and doctrinal restructuring taking place. As the Roman Catholic Church began to weaken and show favouritism towards the Portuguese Crown, King D. Manuel took on a more prominent leadership role within the Church. In 1501, he was granted the right to nominate bishops through "presentation" and oversee their resignation process. During this time, Afonso de Albuquerque conquered Malacca in 1511, Diogo de Azambuja took Safim, D. Jaime, Duke of Bragança, the fortresses of Azamor in 1513, and Jorge Álvares made the first contact with Southern China on Tamão Island (now Hong Kong) near Guangzhou in 1513. A modern state was being developed, accompanied by peace in Europe and maritime expansion. At the same time, the king freed himself from his obligations to the nobles. Maritime commerce helped Portugal to become more prosperous. In the field of the arts, painting moved away from the sacred and began to embrace profanity, while classical forms were revived and reinterpreted. During the reign of D. Manuel, Portuguese painting was influenced by Flemish art due to commercial and artistic relationships with Brussels, Bruges, and Antwerp. The Flemish Book of Hours is highly relevant to the iconographic program of the mural paintings discovered in the St. Leocadia Church. There appear to be similarities among the Altarpiece of the Guild of Antwerp Carpenters (1511), created by Matsys, the Panels of St. Vincent, and the mural paintings found in the apse of the same church [20]. The first piece of work portrays St. John the Baptist and St. John the Evangelist together. This is because in the Middle Ages and Renaissance, it was common to associate these two figures in iconography. St. John the Baptist represents the link between the Old and New Testament, while St. John the Evangelist, symbolized by the royal eagle, represents the New Testament. The holy deacon appears twice in the Panels of St. Vincet. In the mural paintings of the apse of St. Leocadia Church, the symbolic narrative follows the principle of duality in theology. This duality is represented by the warrior vs. the druid and strength vs. traditional wisdom. This concept of dualism dates back to the Iron Age [35].

The mural paintings discovered in the apse of St. Leocadia Church were created during the diocese of D. Fernando de Meneses Coutinho, while the church was under his patronage from 1511 to 1513. Later, the church was reintegrated under the patronage of D. Jaime, Duke of Bragança, by D. Manuel I. The influence of Nuno Gonçalves' painting can be observed through D. Fernando de Meneses Coutinho, who was associated with the circles of power of the Avis dynasty. He was appointed as the head chaplain of Paço da Ribeira in 1514, during the reign of D. Manuel, and held the position while he was the bishop of Lamego [36].

4.1.2. Oil Paintings of Nuno Gonçalves

In the main chapel of the Cathedral of Lisbon, there is a large painting altarpiece that has been dedicated to St. Vincent since the end of the 15th century. The altarpiece remained in its same place in 1545 and 1567 [37]. According to Vítor Serrão, the "Panels of St. Vincent" is a grand ex-voto honouring St. Vincent the Martyr, the deacon-patron of the City of Lisbon, the Kingdom and the Conquests of North Africa. The ex-voto is represented through a set of altarpiece panels that depict the court, nobles, and canons of the cathedral's chapter, friars of two or three religious orders, people in trades and public administration, fishermen, merchants, and manual labourers. These panels (Panel of the

Friars, Panel of the Fishermen, Panel of the Prince, Panel of the Archbishop, Panel of the Knights, and Panel of the Relic) were originally part of the altarpiece in the main chapel of Lisbon Cathedral. The altar of St. Vincent, which contains the venerated tomb and the relics of the saint, was located there. The panels were created between 1460 and 1470, based on historical documentation references, and the style, technique, and compositional elements were confirmed by laboratory examinations. The project was led by the painter Nuno Gonçalves, who was highly praised by Francisco de Holanda and served as the royal painter for D. Afonso V from 1450 onwards [38]. The six Panels of St. Vincent are one of the most important and renowned paintings in Europe from the Quattrocento period. It depicts the glorification of the Holy Infant by St. Vincent, the holy deacon who appears twice in the centre of the frieze as a sign of veneration [39] (Figure 5). The panels depicting the court of D. Afonso V are not just a simple meeting but rather convey the idea of a determined will towards a maritime enterprise, guided by the ideal of the Crusade. According to some authors, the main focus of the panels is to commemorate the captivity of D. Fernando, who failed to conquer Tangier in 1437 [40], as well as of Infante D. Pedro, who fell in the Battle of Alfarrobeira. According to Godinho's historical approach, the reference made to St. Vincent could be traced back to an event that took place on January 1446, perhaps even on the 22nd, which was the date of the feast at St. Vincent. On this day, D. Pedro, Duke of Coimbra, abandoned the regency, and his nephew and future son-in-law D. Afonso V came of age. The dendrochronology of the panels from 2001 coincides with this date [41,42]. The same theme, which is the battle of Alfarrobeira, is also present in the mural paintings of the apse of the St. Leocadia Church and in the fresco at the Paços de Audiência de Monsaraz, which dates back to 1496–1513 [43]. These mural paintings symbolically illustrated both divine and human justice.

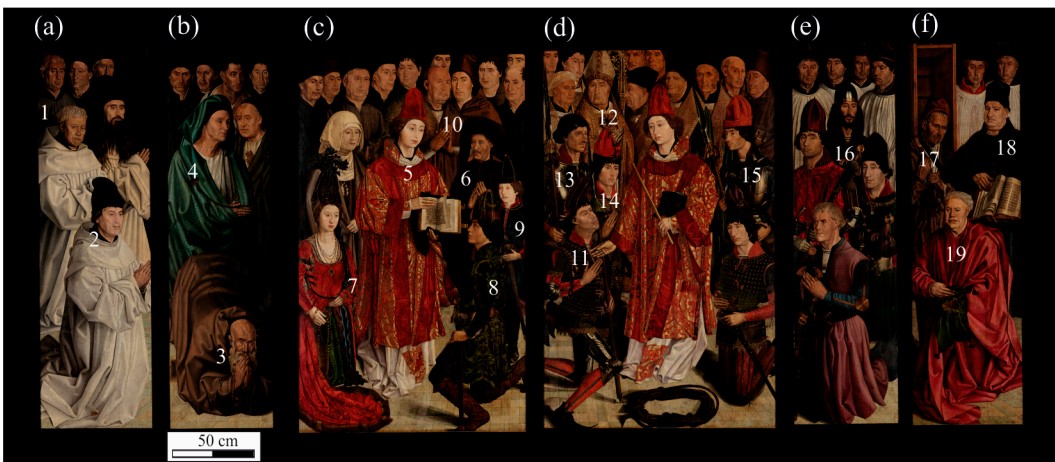

**Figure 5.** Panels of St. Vincent, 1460–1470: (**a**) Panel of the Friars (207.2 × 64.2 cm); (**b**) Panel of the Fishermen (207 × 60 cm); (**c**) Panel of the Prince or Panel of the Infant (206.4 × 128 cm); (**d**) Panel of the Archbishop (206.4 × 128 cm); (**e**) Panel of the Knights (206.6 × 60.4 cm); (**f**) Panel of the Relic (206.5 × 63.1 cm). MNAA, http://www.museudearteantiga.pt, accessed on 16 December 2023. Character identification [37]: 1—John Vincent; 2—Estêvão de Aguiar; 3—Simon Peter; 4—St. John the Evangelist; 5—St. Vincent; 6—King D. Duarte; 7—Queen D. Isabel; 8—King D. Afonso V; 9—Prince D. João; 10—First Duke of Bragança; 11—Infante D. Pedro; 12—Archbishop of Lisbon; 13—Álvaro Vaz de Almada, 14—D. Jaime; 15—Constable Pedro; 16—Infante D. Fernando; 17—Beggar; 18—Jewish rabbi; 19—Friar João Alvares.

As a result of the call for combined efforts by researchers to clarify the identification of the characters in the Panels, this study proposes an interpretation of the work in the context of the Renaissance and dynastic crisis in Portugal during the 15th and 16th centuries, based on the mural paintings of St. Leocadia Church. There were identified some characters from the Vicentine Panels whose features resemble those of the characters in the mural paintings

of the St. Leocadia Church. The following characters have been identified (Figure 5) based on the work of P. Freitas and M.J. Gonçalves [37].

In the Panel of the Friars, (1) Master John Vicent, founder of the Order of St. John the Evangelist [37]; (2) D. Estêvão de Aguiar of the Cistercian Order, the Abbot of the Monastery of Alcobaça (1431–1446) [37,41], who took a stand in favour of the Duke of Coimbra. In the Panel of the Fishermen, (3) Simon Peter, as Armando de Sousa Gomes identified the Franciscan friar [37]; (4) St John the Evangelist, as the character who stands immediately behind the kneeling Franciscan (St. Peter) [37]. In the Panel of the Prince, (5) St Vincent; (6) King D. Duarte, father of D. Afonso V, was identified by D. Markl, following Paviot [41]. His brother, Prince D. Fernando, was held captive in Tangier after an unsuccessful military expedition and later died in Fez in 1443 [33]. (7) Queen D. Isabel, of Portugal [37,38], daughter of the Infante Regent D. Pedro de Portugal, wife and cousin of D. Afonso V, and mother of Princess St. Joan and D. João II. After giving birth in May to prince D. João II, the queen died in December at the age of twenty-three: there was talk of poisoning. (8) King D. Afonso V [44], the king; (9) D. Prince João, future King D. João II; (10) D. Afonso, First Duke of Bragança [45]. In the Panel of the Archbishop, (11) Infante D. Pedro, Duke of Coimbra (son of D. João I, uncle and father-in-law of D. Afonso V); (12) D. Afonso de Nogueira, Archbishop of Lisbon [36,37]; (13) D. Álvaro Vaz de Almada, who took part in the taking of Ceuta, was in Tangier, was captain-general of the royal fleet, mayor-general of the castle of Lisbon, and Count of Avranches in Normandy, and died in battle in Alfarrobeira. (14) Future Cardinal D. Jaime, the youngest son of D. Pedro; (15) Constable Pedro, future King of Aragon [41], the eldest son of D. Pedro. In the Panel of Knights, (16) Infante D. Fernando (Holy Infant). In the Panel of the Relic, (17) the figure of a beggar, symbolising the Holy Infant's spirit of almsgiving and charity; (18) a Jewish rabbi, who is dressed in black, with a radiant six-pointed red star on his chest, holds a book open, showing that the pages turn from left to right, which the painter knew but did not know the Hebrew letters [41]; (19) Friar João Alvares, in the red attire of a notary from Paço da Ribeira, who wrote the Chronica dos Feytos, and Vida e Morte do Infante Santo D. Fernando que Morreu em Fez [46,47]. He was prior of the palace and chronicler of the Holy Infant. According to the opinion of some authors, Friar João Alvares was at the service of the Infante Santo, D. Fernando, as clerk of the Chamber and accompanied him on the tragic expedition to Tangier in 1437 and then during his captivity in Fez. He was rescued by Infante D. Pedro, five years after his death. He later returned to North Africa to bring the relics of the Infante Santo, which currently lie in the Monastery of Batalha [46–48]. In the Vicentine panels, he is showing the relic of St. Anthony that the Duke of Coimbra brought from Padua [41].

The panels representing St. Peter and St. Paul (1465–1490) undoubtedly belong to the same workshop of Nuno Gonçalves and could have been part of the predella of a large altarpiece (Figure 6). The magnificent painting of Santa Joana Princesa (mentioned later in this article), kept in the Aveiro Museum, could perhaps date back to 1471, or shortly before. It was painted by an anonymous author and followed the principles of the Portuguese school of the royal painter Nuno Gonçalves. Her physiognomic features are similar to her brother's ones, the future king D. João II, as he appears in the Panel of the Infant [37]. Infanta St. Joan was a daughter of King D. Afonso V and Queen D. Isabel of Portugal. The Infanta's firmly announced her choice of a religious life, expressing it after the arrival of her father from the capture of Africa, a period during which she exercised, in fact, the regency of the kingdom. Princess St. Joan cared, in the Monastery of Jesus of Aveiro, for the illegitimate son of D. João II, D. Jorge de Lencastre. The devout aunt took great care in the eloquent and pious education of her nephew [49].

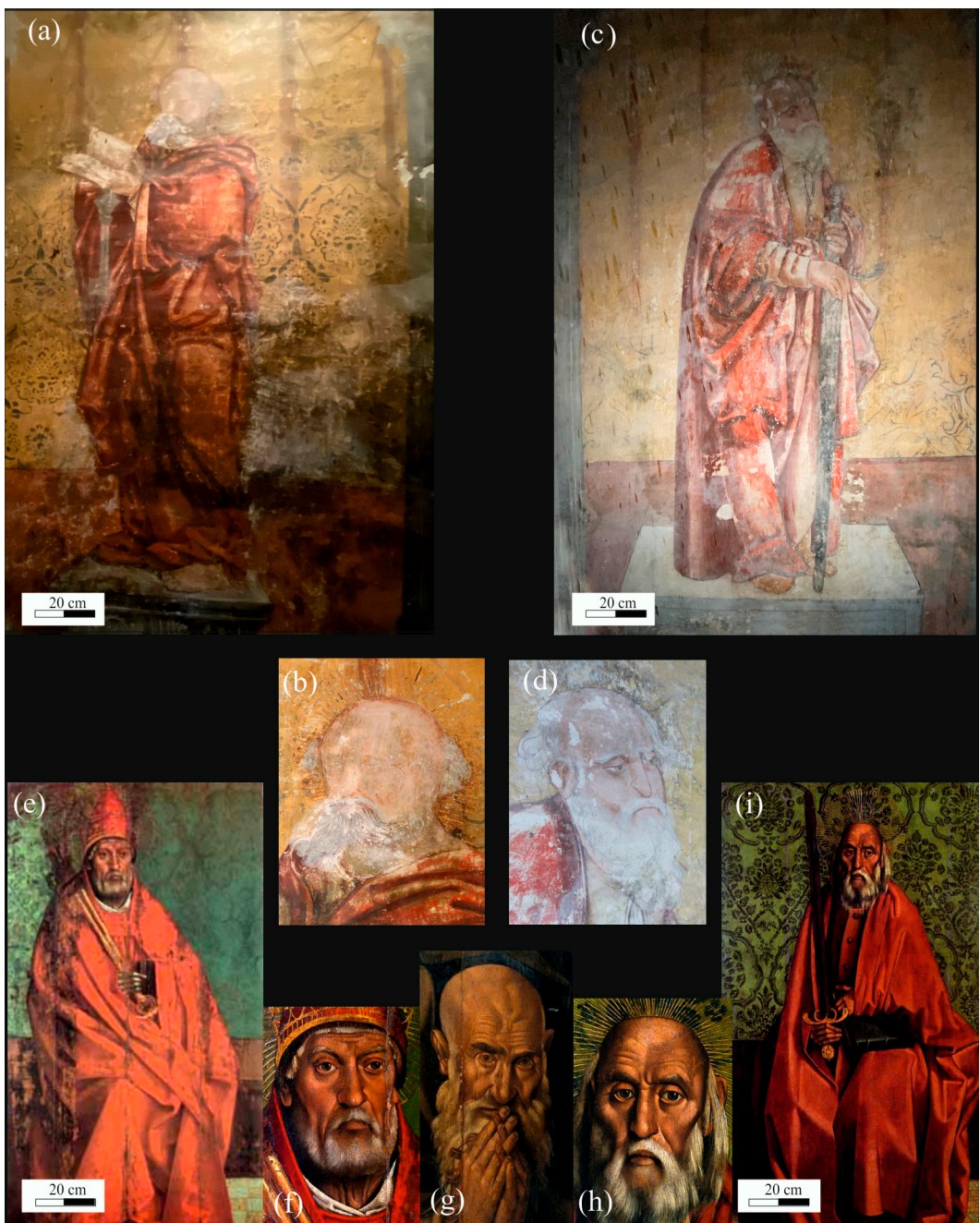

**Figure 6.** The mural paintings of the back wall of the apse depict, on a damask background, St. Peter on the left, holding the gospels and the key to the church, with St. Paul on the right, wielding a sword: (**a,b**) St. Peter (175 × 148 cm) and (**c,d**) St. Paul (175 × 148 cm). Oil painting of Nuno Gonçalves: (**e,f**) St. Peter, 1450–1490 (135.8 × 80.3 cm); (**g**) old Franciscan of the Panel of the Fishermen 1470 (207 × 60 cm); (**h,i**) St. Paul, 1450–1490 (137 × 84 cm).

### 4.1.3. The Mural Paintings from the Apse of St. Leocadia Church

In the Epistle side and Gospel side of the triumphal arch of the apse of St. Leocadia Church, there is a bipartite composition of an inscription, which indicates that the commissioner of the mural paintings was D. Fernando de Menezes Coutinho, in 1511–1513 (Figure 1a,b). It has two shields, one on each side of the opening to the nave—enlarged—and the inscription must be read continuously, without interruption in its five lines of text. "Esta obra ma[ndou]//fazer o egregio senhor//dom fernão [couti]//nho do conce[lho] dellrey e abade//de Santa mar...//maria de moreiras e samta loca[ia]//locaia e adaia[m]//...m

de coimbra e protonotairo [da] See apostollica. —" two lines below, in the right corner, "[tr]as dom".

If we compare the Apostles of St. Leocadia Church (Figure 6a–e) with those painted between 1465 and 1490 by Nuno Gonçalves [49], the resemblance is remarkable. On the other hand, St. Peter also has some features of the old Franciscan in the Panel of the Fishermen. It is interesting to note that in 1927, Sousa Gomes identified this Franciscan friar with Simon Peter [37].

On the Gospel side of the wall, near the old NE angle, one can see the fundamental artistic trends of the Portuguese Renaissance as well as the Flemish (Rogier Van der Weiden) and German (Jakob and Hans Srub) ones. These trends are characterized by classicism and noble realism, which bring life to the dialogue between St. Elizabeth and the Virgin of the Visitation (Figure 7). However, the baroque altarpiece has partially covered Our Lady. There are similarities between the face of St. Elisabeth, who was the mother of St. John the Baptist (Figure 7a,b), and D. Isabel of Portugal [37] (Figure 7c), mother of D. João II, who is portrayed in one of the Panels of St. Vincent, the Panel of the Infant, kneeling at the feet of the saint [41]. Though her figure seems fragile, her presence is strong. D. Pedro, the Constable, brother of Queen D. Isabel, describes his sister as an example of perfection and virtuosity [50].

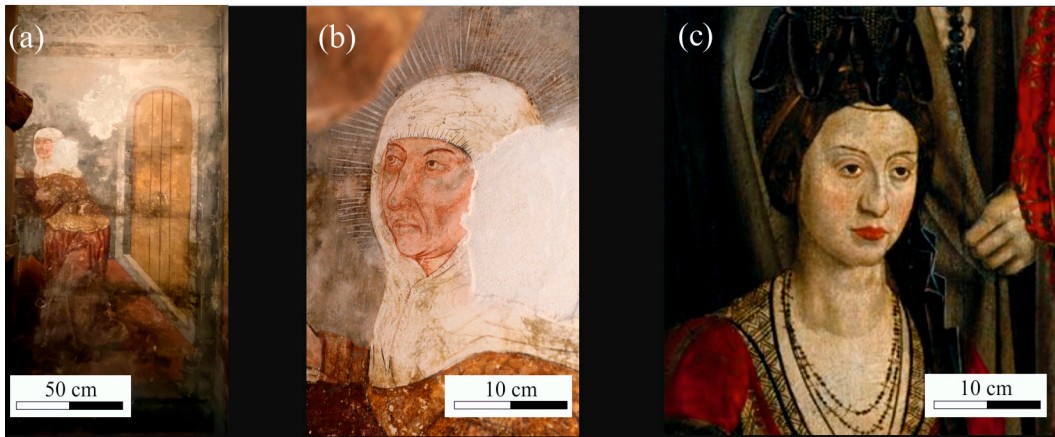

**Figure 7.** The Visitation mural painting (175 × 175 cm) on the Gospel wall of St. Leocadia Church: (**a**,**b**) St. Elizabeth. (**c**) Oil painting of Nuno Gonçalves: D. Isabel of Portugal of the Panel of the Infant.

On the Epistle side, there is a wall adjacent to the back wall that shows the Annunciation to the Shepherds (Figure 8a). This artwork exhibits two main artistic styles, those of Flemish and Spanish Late Gothic. An artist from Bruges created the Illumination of the Book of Hours of D. Duarte between 1401 and 1433 [51], and an oil painting from 1470–1500, currently housed at the Soumaya Museum in Mexico City, both depict a shepherd playing bagpipes, which was a popular musical instrument in northeastern Portugal. The painting is considered to be part of the Portuguese Renaissance period, as the pose and facial features of the shepherd are similar to those of the Infante D. Pedro. The Infante D. Pedro (Figure 8a,b) was a well-built, thin figure with light eyes, and curly reddish hair. He can be seen kneeling at the feet of St. Vincent, holding a military command baton, and standing over the heart (rope) of justice, as portrayed in the Panel of the Archbishop by Nuno Gonçalves. The Infante D. Pedro donated the house of St. Eloi in Lisbon to the clerics of the Order and Rule of St. John the Evangelist [41]. Additionally, in the apse, there is an association of the angel of the Annunciation with St. John the Evangelist. According to A. S. Gomes, the upper left corner of the 15th-century Panel of the Friars depicts Master John Vicent, founder of the Order of St. John the Evangelist. In the Panel of the Fishermen, the character standing immediately behind the kneeling Franciscan (St. Peter) is identified as St. John the Evangelist by the same author [37]. In the mural painting, there is a symbolic relationship between Infante D. Pedro and Master John Vicent. It is not possible to identify

the personality of the other shepherd from the fresco in the church, as it has been partially destroyed by the high altar. There is a proposed similarity between the face of the angel of the Annunciation to the Shepherds and the right side of St. Vincent's face of the Panel of the Infant, which is mentioned in Figure 8a,c. However, while St. Vincent holds the text of the Gospel of St. John/Mass of the Holy Spirit in his left hand [41], in the mural painting, the angel is holding a phylactery which reads "ALEG (..)". This inscription is an extract from the *Book of Isaiah* where the prophet Isaiah predicted the coming of the Savior and a king called Emmanuel. The passage from Luke 2:14 states that the army of angels rejoices because eternal salvation has appeared to the human race. In the Vicentine panel, one can notice a coiled rope that has been thrown to the ground. This attribute of the saint is quite rare and is linked to a miracle that occurred to Portuguese sailors who went to look for his body in Valencia [41]. In the mural painting of St. Leocadia Church, the rope gives way to a shadow of concentric dark brushstrokes, and the shepherd with a bagpipe is kneeling by the riverbank.

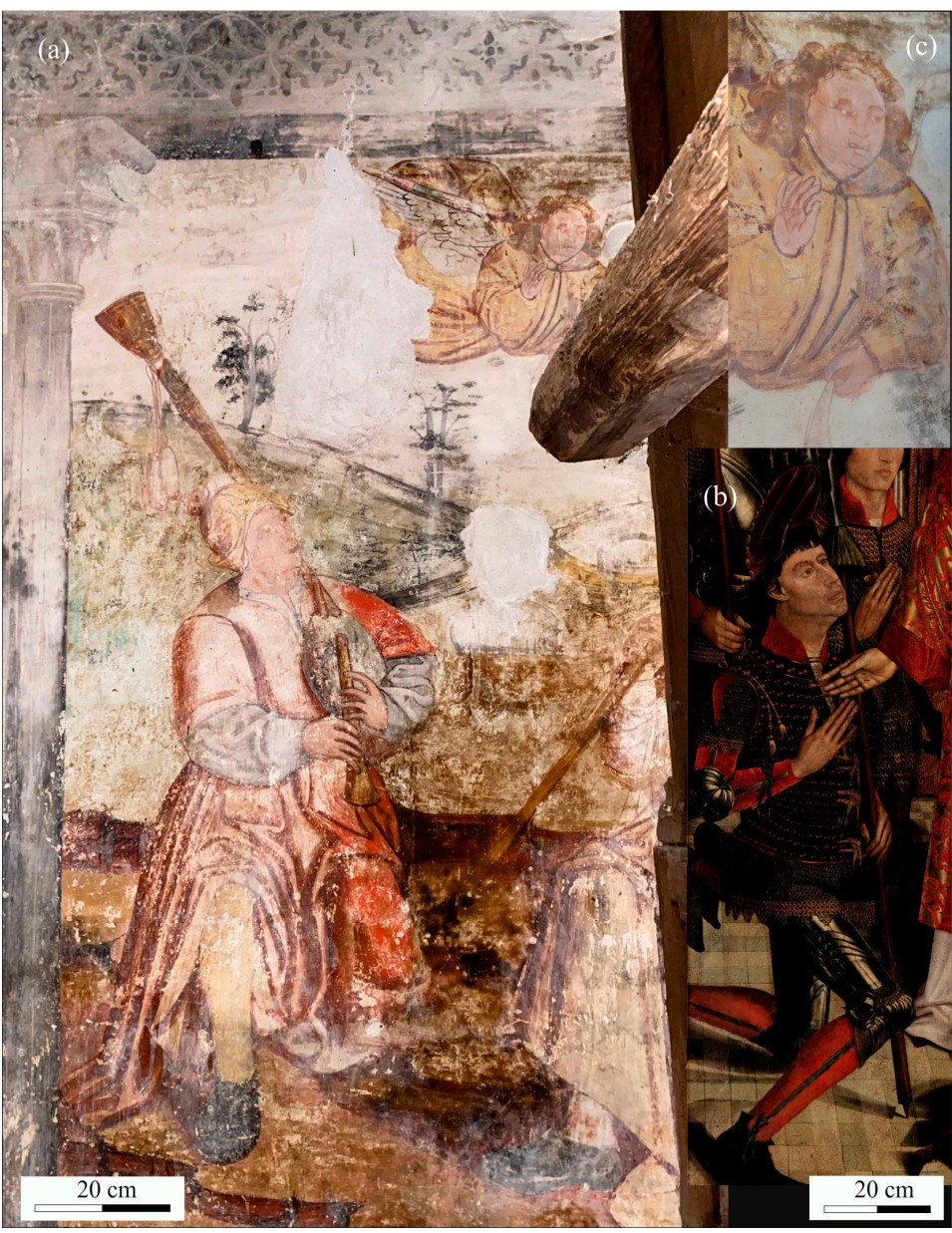

**Figure 8.** Mural painting of the Epistle wall of the apse of St. Leocadia Church: (**a**) The Annunciation to the Shepherds (175 × 175 cm). Oil painting by Nuno Gonçalves: (**b**) Panel of the Archbishop; (**c**) the angel of the Annunciation, in detail.

The continuity of this mural painting, on the wall of the apse on the Epistle side, displays Jesus' Presentation in the Temple (Figure 2b) (the presentation of Jesus and the purification of the Virgin was held forty days after Jesus' birth in Bethlehem), which shows Germanic Renaissance influences, as can be seen from the engraving by Wolgemut from 1490 and the oil painting by Holbein from 1500. Here, we can see the Archbishop of Lisbon in 1459, identified as D. Afonso de Nogueira from the Panels of St. Vincent. By observing the mural painting of the Church, despite its poor state of repair, three supporters of the Infant D. Pedro can be found behind the Archbishop (as in the Vicentine Panels): his sons D. Jaime and D. Pedro, the Constable; and D. Álvaro Vaz de Almada, his "brother in arms" (Figure 2e).

On the same wall on the Epistle side of the apse, over the gap that opens onto the sacristy, there is a commemoration of the Flight into Egypt, which occurred some two years after the birth of Jesus (Figure 9b). In this painting, which is now barely noticeable due to its poor condition, St. Joseph appears to be wearing the clothing of a pilgrim. Our Lady carries the baby Jesus on her lap and is transported by a donkey which is pulled by St. Joseph with a rope. This painting was influenced by the French Gothic style and an engraving by Wolgeman from 1491 (Figure 10e). The face of St. Joseph of the mural painting of St. Leocadia Church is identical to that of the Franciscan kneeling with his elbows on the ground praying with a rosary, which is depicted in the Panel of Fishermen, from the Panels of St. Vincent. He has the same expression of suffering. Our Lady's face is similar to that one of Infanta St. Joan, painted by Nuno Gonçalves, in 1472–1475 (Figure 9c) [52,53]. In the Memorial of the Monastery, Soror Margarida Pinheira or Soror Catarina Pinheira, wrote in what is a true biography: "a very sharp face and body, a very graceful forehead, very beautiful green eyes, a medium and good-looking nose, a thick and rebellious mouth, a round face, a very beautiful throat and hands, more than one could find and see in no other woman, tall and large with a straight body, very fit and elegant, the figure is a representation of a great lady and state" [54].

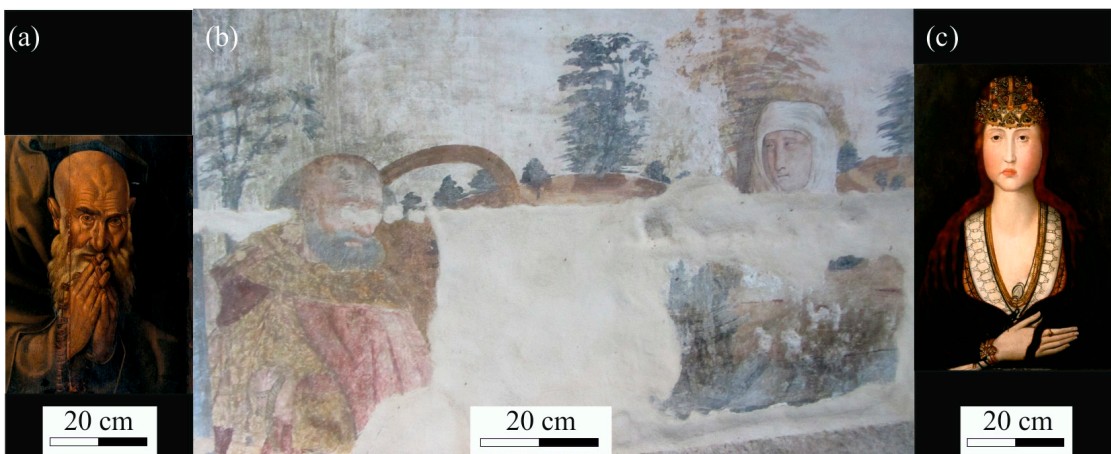

**Figure 9.** Mural painting of the Epistle wall of the apse of St. Leocadia Church: (**b**) The Flight into Egypt. Oil paintings by Nuno Gonçalves: (**a**) Franciscan of the Panel of the Fishermen; (**c**) Santa Joana Princesa (portrait of Princess St. Joan,1472–1475, oil painting on oak wood, 60 × 40 cm); http://www.matriznet.dgpc.pt, accessed on 16 December 2023.

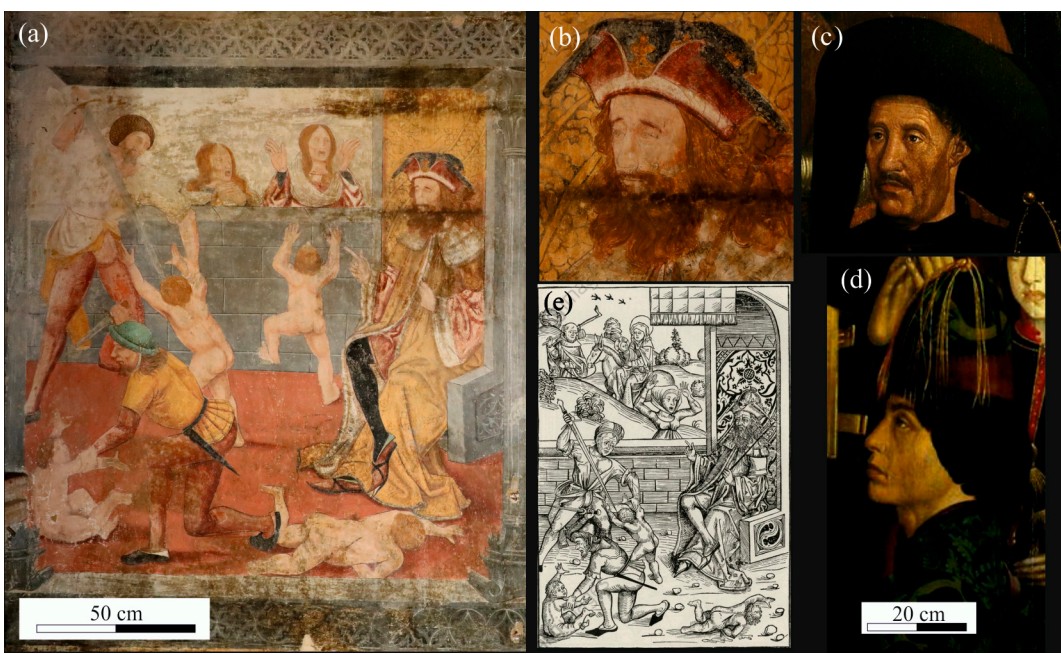

**Figure 10.** Mural painting of the Gospel wall of the apse of St. Leocadia Church: (**a**,**b**) The Slaughter of the Innocents. Oil painting by Nuno Gonçalves, Panel of the Infant: (**c**) King D. Duarte; (**d**) King D. Afonso V. (**e**) Michael Wolgemut Woodcut, 1491, https://www.akg-images.com/asset-management, accessed on 1 January 2020.

On the wall on the Gospel side, next to the triumphal arch, the mural painting depicts The Slaughter of the Innocents (Figure 10a), based on the same engraving by Michael Wolgemut from 1491 [55]. It reflects the influence of the Late Gothic (late 15th–early 16th centuries) mural painting with the same theme, in the Gospel wall of the apse of the Cathedral of Our Lady of the Assumption in Mondoñedo, Lugo Province, Galicia. The three-quarter face of the character who embodies King Herod matches the three-quarter face and profile of two figures of the Panel of Infant by Nuno Gonçalves, which have been identified as the Kings D. Duarte and D. Afonso V (Figure 10b,c). King Herod is remembered for his cruelty, including the murder of three of his sons. D. Duarte, King of Portugal, was also unwilling to accept the ransom conditions, which required the return of Ceuta in exchange for the surrender of his brother, Infant D. Fernando. The death of Prince D. Fernando was a collective sacrifice, and he was sacrificed like an innocent man. In the Vicentine panel, D. Duarte is the iconic man wearing a chaperon. Just as King Herod ordered the execution of the new-borns in Bethlehem for fear of losing the throne to the "King of the Jews", D. Afonso V, King of Portugal, and son of D. Duarte, is associated with the idea of being a "Bad Judge" [20,43]. Particularly for letting himself be negatively influenced by his uncle, the Count of Barcelos, who later became the First Duke of Bragança, D. Afonso, half-brother of Infante D. Pedro. Therefore, the king was convinced that his uncle and father-in-law, Pedro, intended to usurp the throne. As a result, the armies of D. Afonso V confronted those of D. Pedro in the Battle of Alfarrobeira in 1449, where the latter died in combat. The other personage who is found on the top left corner of the mural painting is identified, according to some authors, with D. Afonso, First Duke of Bragança of the Panel of the Infant. While the Duke of Bragança in the St. Leocadia mural painting is depicted in the age he would have been at the time of the Battle of Alfarrobeira, i.e., 72 years old, in the Vicentine panel, he must have been the age he was at the end of his life. With this mural painting, D. Manuel's intention was to demonstrate that he considered D. Afonso V not to have the profile to be a fair king, unlike him. Diogo Lopes Rebelo, master of D. Manuel, prior of Clermont and professor of Theology in Navarre, defended justice, as the virtue most necessary to the king: «Justice and mercy should be appropriate to the king in relation to his subjects, distinguishing two axial effects in justice: the first is not to

harm anyone and the second is to prevent usurping what belongs to others. The opposite constitutes injustice. In this way, it is up to the king to administer justice, always asking God for it and exercising it with diligence and mercy: the king uses justice, mixing it with mercy, clemency, and kindness, and tends more towards salvation than condemnation» [20,56]. After the death of D. João II and the restoration of the House and Dukedom of Bragança, in 1496, by a letter of donation from King D. Manuel I, D. Jaime inherited the Dukedom.

Still on the same wall of the apse, close to the high altar, the mural depicts Jesus among the Doctors (Figure 2a). Not only does the Renaissance influence of Durer (1506) prevail here, but also the inspiration of the painting of Matsys (1509–1511) and of the French School of the 16th century, as well as the iconography and symbolism of the polyptych of St. Vincent Outside the Walls. Jesus, a teenager, attracts all the doctors of the Temple with his intelligent questions. In the images that illustrate this scene, several similar details can be observed, such as the central and highest position of Jesus, with long blond hair, gesturing with his hands while talking to the doctors in awe of that child. A correlation of similarity is made to the teenager, the future king D. João II from the Panel of the Infant. Although we cannot recognise the facial features of Jesus of the mural painting of St. Leocadia Church, we can identify some similarities between the two teenagers, such as their height and the gesture of their left hand. Among the characters who surround Jesus in the mural painting of St. Leocadia, others are also recognizable, such as the figure of the Cistercian of the Panel of Friars, D. Estevão de Aguiar (in a black cap) (Figure 2c). Behind him is a personage in a green cap, who seems to correspond to the Jew of the Vicentine Panel of the Relic, who also has a green hat. We also see that the figure of the Panel of the Relic, identified as João Alvares (Figure 2d), in red attire, is holding a closed book in his left hand, and is located on the right side, next to Jesus, as he was one of the supporters of the Regency of D. Pedro. Behind this, there is "the pilgrim", who is in front of an open coffin in the Panel of the Relic. The "pilgrim" symbolizes the various routes that the remains of Infante D. Pedro had to follow until they were laid to rest in his tomb in the Monastery of Aljubarrota.

*4.2. Experimental Campaign*

Table 1 shows the linear relationship between the original measurements and the ratios for each pair of images: the ratio of the distance between the corners of the eyes and the distance between the middle of the eyebrow and the tip of the nose; the ratio of the distance between the corners of the eyes and the distance between the middle of the eyebrow and the top of the mouth; the ratio of the distance between the corner of the eye and the corner of the mouth and the mouth length; and the ratio of the distance between the middle of the eyebrow and the base of the eye and the inter-pupillary distance.

**Table 1.** Facial feature original measurements. Calculation of the ratios.

| Image Code | Original Measurements | | Ratios | | All Values | |
|---|---|---|---|---|---|---|
| | E | D | E | D | E | D |
| FR_70_ASP | 1 | 0.912 | 1 | 0.702 | 1 | 0.946 |
| SP_VL | 1 | 0.983 | 1 | 0.914 | 1 | 0.984 |
| R_S_V2 | 1 | 0.989 | 1 | 0.914 | 1 | 0.992 |
| DP_PF | 1 | 0.981 | 1 | 0.978 | 1 | 0.963 |
| FR_79_FRC | 1 | 0.966 | 1 | 0.914 | 1 | 0.977 |
| FR_70_SL | 1 | 0.975 | 1 | 0.993 | 1 | 0.985 |
| DD_CH | 1 | 0.963 | 1 | 0.961 | 1 | 0.983 |

Legend (See Figure 4 for details): FR-70_ASP_E vs. D: Franciscan friar vs. St. Peter; SP_VEL_E vs. D: St. Paul vs. St. Paul; R_SV2_E vs. D: Queen Elizabeth vs. St. Elizabeth of the Visitation; DP_PF_E vs. D: Infante D. Pedro vs. the Shepherd of the Annunciation; FR_79_FRC_E vs. D: Franciscan friar vs. St. Joseph of the Flight into Egypt; FR_70_SL_E vs. D: Princess St. Joan vs. Our Lady of the Flight into Egypt; and DD_CH_E vs. D: King D. Duarte vs. the man with a chaperon of the Slaughter of the Innocents.

The graphs in Figure 11 show the degree of influence of the selected parts of the face on recognizing the similarities between the images of the Nuno Gonçalves oil painting and the images of the mural painting of the Church of St. Leocadia; facial feature specific models make the facial recognition process possible. The graphs represent the computed percentage similarities between two given images of seven known individuals.

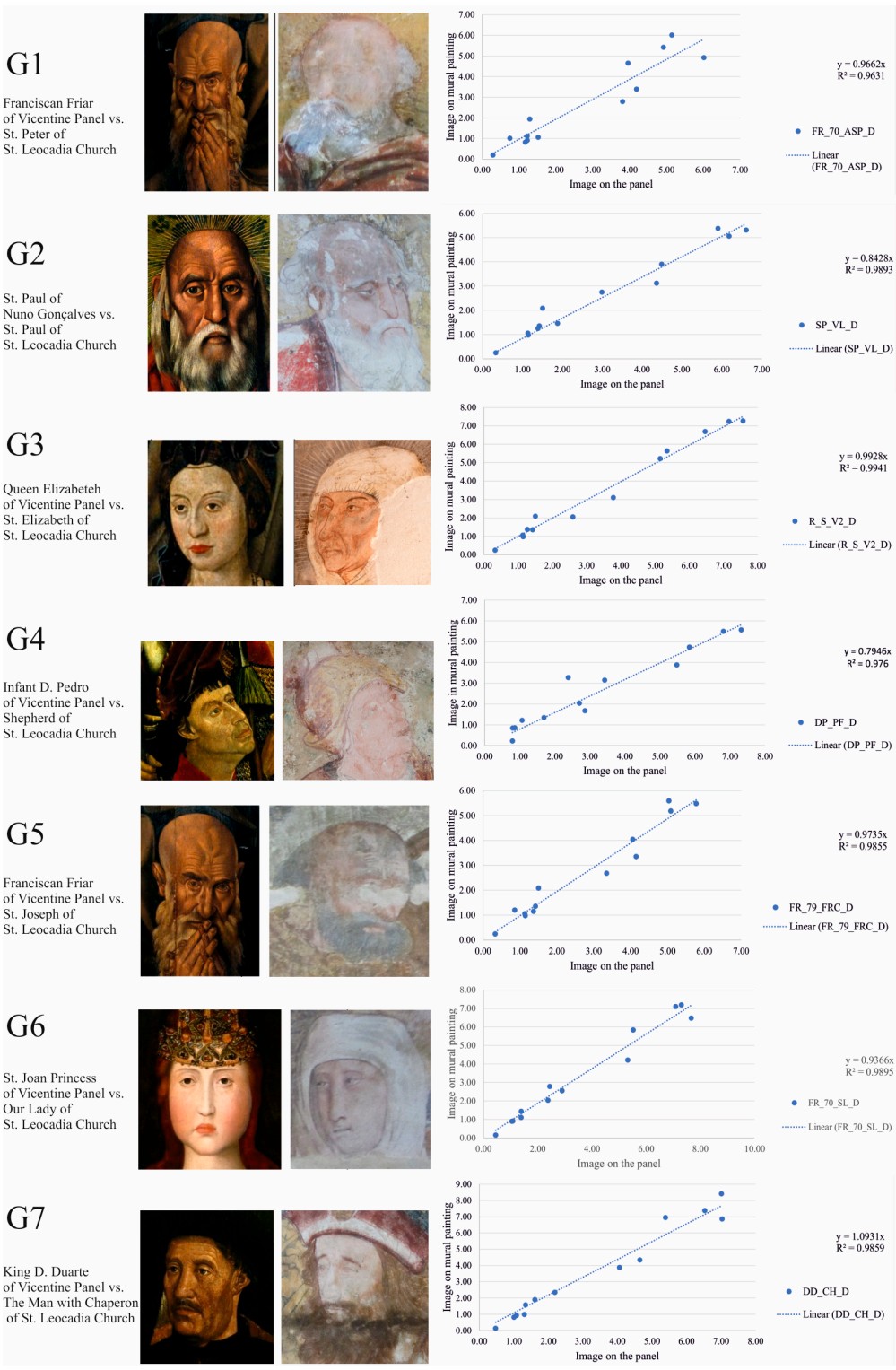

**Figure 11.** Graphs expressing the linear regression equation without constant for each pair of values (image pairs).

## 5. Discussion

The iconographic interpretation of those mural paintings aims to understand the function and meanings of depictions and the reasons behind the Manueline state's commission for those portraits and the development of rituals for their royal ancestor worship throughout the Avis dynasty. It examined the correlation between oil paintings by Nuno Gonçalves, including the collective portraits of the Panels of St. Vincent, and the characters of the mural paintings of St. Leocadia Church. The subject matter of the paintings expressed the royal family's hope for the eternal life of their ancestors and the prosperity and regeneration of the nation. The portraits found in the mural paintings appear as historical paintings incorporated into the religious action itself. The faces of the king, queen, princes, and other important figures of the 15th century are placed in the mural paintings of St. Leocadia Church with the aim of placing the royal family under divine protection and, at the same time, showing their devotion.

The type of portrait of monarchs and princes in St. Leocadia Church seems to have been taken from the portraits of Nuno Gonçalves, as this research has shown. In the 16th century mural paintings, the representation of concrete characters is inserted into the sacred representations themselves and even into the representation of saints. Here, historical painting and the portraiture of a family dynasty are identified with the need to perpetuate memory and valour as a Renaissance ethos.

The figures are represented using a three-quarter view traditionally used for images of Jesus, Our Lady, saints, and virtuous men and women. The mural paintings presented them as royal figures, possessing the moral authority appropriate to ruling what they regarded as the just state in the world, contrasting with other ones, who carried out a disastrous national policy that immolated innocents on the altar of the nation, or who were considered "bad judges". With these mural paintings, St. Leocadia Church turned into a memorial site of the natural parents of King D. Manuel. He used the power of images to elevate the status of their low-ranking natural parents and their royal authority. The images of kings and princes as saints or the family of Jesus signify humility, wisdom, and the obligation of Christian filial piety. St. Leocadia Church, in this way, was transformed into an official space to reinforce royal legitimacy and authority.

An experimental campaign was carried out, through a direct facial comparison between pairs of images of Nuno Gonçalves and mural paintings of St. Leocadia Church: Franciscan friar vs. St. Peter; St. Paul vs. St. Paul; Queen D. Isabel vs. St. Elizabeth of the Visitation; Infante D. Pedro vs. the Shepherd of the Annunciation; Franciscan friar vs. St. Joseph of the Flight into Egypt; Princess St. Joan vs. Our Lady of the Flight into Egypt; and King D. Duarte vs. the man with a chaperon of the Slaughter of the Innocents. The result was a match between 96% and 99%, which means that they are considered practically identical. In the facial recognition programme exercise, a match superior to 75% is considered an identity, and therefore there is a strong probability that there was an important influence of the Nuno Gonçalves painting models on the mural paintings in St. Leocadia Church.

From the composition of surfaces arises that exquisite harmony and grace in bodies and faces which is called beauty. With the anatomical distances' ratios and the analysis of the pairs of painted historic faces and bodies, it was possible to find the following portrait painting practices:

(1) The forehead, nose, cheeks, and chin of each pair of faces were found on the same parallels;
(2) It was possible to establish the similarity between the contours, lights, and shadows of the faces and bodies;
(3) It was possible to find real portraits, well-defined faces, and in some of them, the ageing process, as in St. Elizabeth, by comparing her to Queen D. Isabel;
(4) The movements and poses of the old men correspond to tired physiognomies with similar sunken surfaces, wrinkles, and dry hands due to thinness; they are sad people, whom worry afflicts and thoughts harass, as in St. Peter, St. Paul, St. Joseph, and the old Franciscan;

(5) Melancholic men have frowning foreheads and languid heads; all their limbs droop as if they were tired and careless, like the man in the chaperon;

(6) In happy men, the movements are free and with certain pleasant inflections, such as the head directed upwards towards the sky, smiling, like the shepherd of the Annunciation or the infant D. Pedro;

(7) Young people have a physiognomy of soft, gentle lights and shadows; they do not have any roughness or protruding angles, but they have a beautiful, delicate physiognomy, as Our Lady and Princess St. Joan.

## 6. Conclusions

The iconological interpretation of the mural paintings of St. Leocadia Church, especially in the apse, was aided by the fact that they are based on historical events. Literary sources, archive research, and international artistic studies all contributed to invaluable information about the Manueline style. The European and overseas influence in the iconography and materials used in these mural paintings is noticeable, though the symbolic significance of the "St Vincent Panels" was decisive. It seems as if the Portuguese painters learned from external practices, though they adapted them to Portuguese traditions. The iconological programme creates a rearrangement of the architectural space, which was typical of the first Renaissance, obeying criteria with a catechetical purpose, while carrying a sense and meaning of redemptive salvation, alluding to maritime exploration at the same time. The entire world of symbolic values reflects the socio-economic, political, and religious context that characterized Portugal at the time. It also reflects the macro-imperial ideology of D. Manuel, who identified himself with the Emmanuel of Isaiah's prophecies.

The results of the iconographic analysis of the Early Renaissance mural paintings of St. Leocadia Church, presented for the first time, will contribute to a better understanding of the history present in the Portuguese Chroniclers of the 15th and 16th centuries and its relationship with the representation on the Panels of St. Vincent. At the same time, considering the results garnered throughout the study of the Manueline mural paintings in the Church of St. Leocadia, it was possible to propose that are essentially part of an *ethical* Renaissance, different from the Italian *aesthetic* Renaissance.

The forensic facial comparison study conducted in this study, proposed that the characters painted in the mural paintings of St. Leocadia Church were the posthumously produced models of the Quattrocento royal painter. The result, matching between 96% and 99%, means that they are considered practically identical. Deep learning-based facial recognition analysis is useful as an additional tool in works of art with historical importance.

This research is a valuable contribution to the conservation of cultural heritage. It is also important for teachers of the conservation and restoration of historical and artistic works, and for the development of reconstruction and heritage conservation interventions.

**Author Contributions:** Conceptualization, E.S.; methodology, E.S., J.A. and R.M.; validation, E.S., J.A., R.M. and D.M.F.-L.; formal analysis, E.S., J.A. and D.M.F.-L.; investigation, E.S., J.A. and R.M.; resources, E.S., J.A., R.M. and D.M.F.-L.; data curation, D.M.F.-L.; writing—original draft preparation, E.S.; writing—review and editing, E.S., R.M. and D.M.F.-L.; visualization, E.S. and J.A.; formal analysis, R.M.; photographs—D.M.F.-L. and E.S.; supervision, E.S.; project administration, D.M.F.-L.; funding acquisition, D.M.F.-L. Composition of an Inscription in the Triumphal Arch of the apse of St. Leocadia Church, by R.M. All authors have read and agreed to the published version of the manuscript.

**Funding:** This work was financed with national funds through FCT—Fundação para a Ciência e Tecnologia, I.P., of Portugal, under the projects with the references CEECIND/03568/2017, [UIDB/00073/2020], and [UIDP/00073/2020] of the I, D unit Geosciences Centre (CGEO) of Coimbra University (Portugal). Funding was also provided by national funds through UTAD: www.utad.pt.

**Data Availability Statement:** Data are contained within the article.

**Acknowledgments:** The first author would like to thank the researcher David Martin Freire-Lista for the facilities provided to write this manuscript. We also thank some photographs by Yanira Peña Gabriel, which were used in this study.

**Conflicts of Interest:** The authors declare no conflicts of interest.

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
