# Peer review of "Proto-Early Renaissance Depictions, Iconographic Analysis and Computerised Facial Similarity Assessment Connections: The 16th Century Mural Paintings of St. Leocadia Church (Chaves, North of Portugal)"

_heritage, doi:10.3390/heritage7040096_

Round 1
Reviewer 1 Report
Comments and Suggestions for Authors
The idea at the basis of the manuscript is potentially interesting. However, most of the paper is devoted to the artistic/historical description of the paintings, which is undoubtably important but is treated in the paper in an excessive detail, with the same concepts often repeated several times in the discussion. The most disappointing part of the work is the so-called ‘deep face recognition model’ described by the authors, which is heavily based on the manual calculation of a few anthropometric facial parameters that are compared between pre-selected couples of images which is – as admitted by the same authors – more similar to a a posteriori forensic facial comparison than to a deep-learning based approach. To be considered for publication, in my opinion the manuscript should be completely rewritten, drastically reducing the artistic and historical discussion to the essential needed for understanding the interest of the comparison presented by the authors. In that respect, it should be made clear by the authors that the results presented are based on rather standard forensic procedures, and the difference between this approach and the deep learning techniques, which would have required a much larger database for training, should be made clear. The terms ‘Computerised Face Recognition Connection’ should be substituted in the title with ‘Computerised Facial similarity assessment’, or something similar.
Some of the images shown seem to come from poor quality photographs, evidencing reflections, perspective deformation and shadows. Better quality photographs should be used in the revised manuscript.
Comments on the Quality of English LanguageThe English must be improved, many sentences are difficult to follow and understand. In general, the whole strucure of the manuscript suffers from the lack of clarity of the text.
Author Response
Dear Reviewer,
Thank you for carefully reading our manuscript and for sending comments and suggestions.
The comments are pertinent and allowed us to improve the manuscript.
As it was necessary to carry out an exhaustive and extensive revision of the English, we sent the revised version in the original format and the revised version but clean. In this way, you will be able to check and compare the two versions: original and revised.
We have also updated the material and methods part, as suggested.
Regarding the quality of the photographs, unfortunately, we can't improve all of them. The frescoes are in poor condition and the photos we present are the best we can do. However, around half of the figures were replaced with better quality photos, in the new version of the manuscript.
The pictures of the São Vicente Panels were obtained from the Internet, in the site of Museu Nacional de Arte Antiga (Lisbon), http://www.matriznet.dgpc.pt/MatrizNet/

Reviewer 2 Report
Comments and Suggestions for Authors
The reference to Wolgemut engraving of Massacre of Innocents needs to be specified. Is a woodcut being referred to?
Parts of the article are in perfectly clear English and others (the beginning, for example), are not.
I would drop the reference to Panofsky's Meaning in the Visual Arts, with all due respect for his work.
It seems to me that the gist of the article has to do with these paintings, not with the facial recognition technology at all, which is a supplementary argument. The distinction between the ethical and aesthetic Renaissances needs to be made earlier and further developed. The paintings are variously referred to as late Gothic and old master: they can't be both and look much more nearly late Gothic, verging into Renaissance. The force of the facial recognition argument would be clearer if it was left to the end of the end of the article, after traditional art historical analysis had been fully accomplished. It would then be possible to make clear how the facial recognition analysis makes more definite or actually changes what can be argued about the frescoes. It seems as though more is being claimed for the facial recognition technology than can be justified: that the comparison with the St. Vincent panels may be supported by it, but exists quite persuasively without it.
Comments on the Quality of English Language
As noted above, the language needs to be simplified, condensed, and in some cases made grammatical. The co-authored status is evident in the lurches in the comprehensibility of the prose.
Author Response

(The authors gave the same response as above.)

Reviewer 3 Report
Comments and Suggestions for Authors
The manuscript reports a method to investigate facial similarity analysis in paitings. It was applied to an almost significant number of different personages (around 45).
Despite the goal is clear and well described, the literature cited as reference is poor. More papers about the method and the consolidated or pioneer methods should be added in order to underline the novelty of thier method as well as the progress in their technology.
The most of the manuscript is dedicated to the iconographic comparison. This is necessary of course, but the main core of the mansucript should be the discussion on the results and it is very short, comparing the rest of the text. in addition, the "Historical-Political-Artistic Framework" paragraph occupies the half of the manuscript, so that the reader loose the goal of the study. I suggest to revise the manuscript hilighting the details on the goal and method. The rest could be reported at the end of manuscript, only for the interesed readers.
From the technical point of view, No equations are reported, even if they are cited in the paragraph 4.2. In this way, the description of the method is very poor. In addition, the resolution of the images reporting the XY representation is not good. Errors on each point are not reported. Numbers reported in Table 1 are not specified. Why comma, and not points for the decimal numbers?
The paragraphs 4.2 and 5 should be better detailed.
At this stage, the manuscript cannot be accepted for the publication.
Comments on the Quality of English Language
The quality of english is good.
Author Response

(The authors gave the same response as above.)
